Review Article

# Variant scoring tools for deep mutational scanning

Hasan Çubuk [1,4], Xinyi Jin [2,3,4], Belinda Phipson [2,3], Joseph A Marsh [1✉] & Alan F Rubin [2,3✉]

## Abstract

**Deep mutational scanning (DMS) can systematically assess the effects of thousands of genetic variants in a single assay, providing insights into protein function, evolution, host-pathogen interactions, and clinical impacts. Accurate scoring of variant effects is crucial, yet the diversity of tools and experimental designs contributes considerable heterogeneity that complicates data analysis. Here, we review and compare 12 computational tools for processing DMS sequencing data and scoring variant effects. We systematically outline each tool's statistical approaches, supported experimental designs, input/output requirements, software implementation, visualisation capabilities, and key assumptions. By highlighting the strengths and limitations of these tools, we hope to guide researchers in selecting methods appropriate for their specific experiments. Furthermore, we discuss current challenges, including the need for standardised analysis protocols and sustainable software maintenance, as well as opportunities for future methods development. Ultimately, this review seeks to advance the application and adoption of DMS, facilitating deeper biological understanding and improved clinical translation.**

**Keywords** Deep Mutational Scanning; Multiplexed Assays of Variant Effect; Functional Genomics; Bioinformatics; Software
**Subject Category** Computational Biology

## Introduction

Proteins are the primary effectors of cellular function, translating genetic information into the structural and biochemical activities that sustain life. Variants in protein-coding genes are often implicated in genetic disease, as they can alter protein structure, stability, or interactions, leading to disrupted cellular processes. Deep mutational scanning (DMS) is an experimental technology designed to efficiently and systematically measure the impact of many thousands of protein variants in a single pooled assay (Fowler et al, 2010; Fowler and Fields, 2014). DMS has been applied to a wide range of research areas, including understanding basic protein properties (Olson et al, 2014; Matreyek et al, 2018; Faure et al, 2022; Clausen et al, 2024), elucidating the dynamics of protein evolution (Hietpas et al, 2011; Firnberg et al, 2014; Starr et al, 2017; Mehlhoff

et al, 2020), providing evidence for clinical variant classification (Majithia et al, 2016; Findlay et al, 2018; Mighell et al, 2018; Bridgford et al, 2020; Jia et al, 2021; Gebbia et al, 2024), benchmarking predictive models (Livesey and Marsh, 2020; Frazer et al, 2021; Notin et al, 2023), enhancing rational protein design (Tinberg et al, 2013), and examining host-pathogen interactions (Haddox et al, 2018; Chan et al, 2020; Dadonaite et al, 2024).

A typical DMS experiment includes four key steps: generating a variant library, performing a pooled selection assay, deep sequencing to quantify variants, and variant effect scoring. Variant libraries are mutant pools for the protein of interest created using various molecular biology techniques, including primer-based methods that use short oligos containing user-defined mutations (Firnberg and Ostermeier, 2012; Jain and Varadarajan, 2014), CRISPR-based mutagenesis (e.g., saturation genome editing (Findlay et al, 2014)), and proprietary vendor solutions. The mutants are then assayed based on their associated phenotypes to perturb variant frequencies in the pool over the course of the experiment (Starita and Fields, 2015). These screens may measure protein binding (Fowler et al, 2010; Nedrud et al, 2021) or fluorescence (Reich et al, 2015; Matreyek et al, 2018; Amorosi et al, 2021), but typically use cell growth or survival as a readout (Hietpas et al, 2011; Firnberg et al, 2014; Giacomelli et al, 2018; Ahler et al, 2019), enriching functional genetic variants in the population while depleting detrimental ones. High-throughput sequencing is then used to count observations of every variant in each sample or time point, enabling the calculation of scores summarising the variant effects for downstream analysis and interpretation (Fig. 1) (Fowler et al, 2011, 2014).

Several tools have been authored for processing DMS sequencing data and generating variant effect scores, but the diversity of statistical approaches and software implementations, coupled with the heterogeneity in experimental design for DMS experiments, makes it difficult to compare approaches or decide on the best analysis strategy for a given dataset. Here, we aim to provide a comprehensive review describing and comparing the key attributes and capabilities of each tool, both to inform the decision-making process for researchers generating DMS data and to provide an introduction for experts in statistical methods development, ultimately advancing the field of DMS and its diverse applications.

## Overview of variant scoring tools

In DMS experiments, data analysis is split into two main stages: processing FASTQ files from high-throughput sequencing into counts for each variant for each sample and then using these counts to

[1]MRC Human Genetics Unit, Institute of Genetics and Cancer, University of Edinburgh, Edinburgh EH4 2XU, UK. [2]Bioinformatics Division, The Walter and Eliza Hall Institute of Medical Research, Parkville, VIC 3052, Australia. [3]Department of Medical Biology, The University of Melbourne, Parkville, VIC 3010, Australia. [4]These authors contributed equally: Hasan Çubuk, Xinyi Jin. ✉E-mail: joseph.marsh@ed.ac.uk; alan.rubin@wehi.edu.au

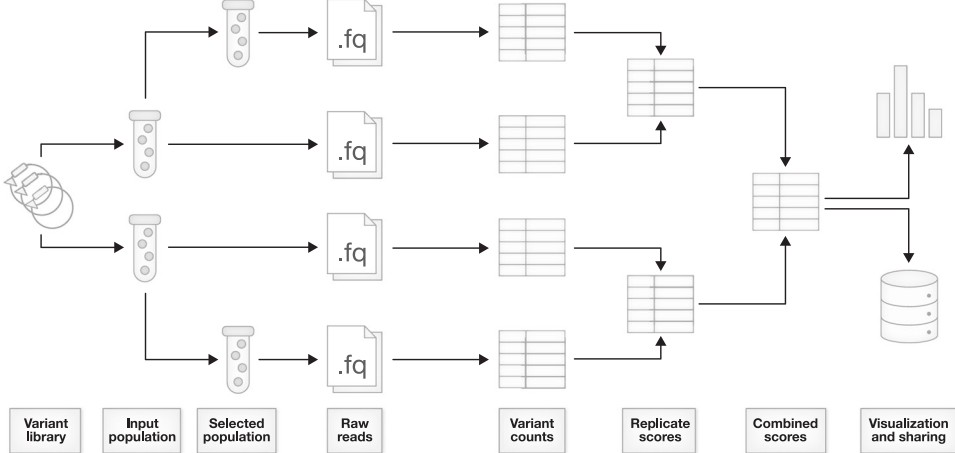

**Figure 1. Schematic of a representative deep mutational scanning (DMS) experiment.**

This shows a DMS experiment with two replicates and two time points (input and selected). Each variant population is sequenced to produce a file containing raw reads, which are then counted and scored. The final scores are then used to generate visualisations, and the final output is shared in a community database or through another mechanism.

calculate variant effect scores. A basic "two-population" experimental design consists of two samples collected at input (pre-selection) and output (post-selection) time points. In its simplest form, a variant effect score $s_v$ is calculated as the log ratio of the variant $v$'s frequency in the output sample determined by dividing the variant count $c_{v,\text{output}}$ by the total count of all variants in the sample, also known as the library size, to its frequency in the input sample:

$$s_v = \log_2 \left( \frac{\left( \frac{c_{v,\text{output}}}{\sum_i c_{i,\text{output}}} \right)}{\left( \frac{c_{v,\text{input}}}{\sum_i c_{i,\text{input}}} \right)} \right)$$

where $i$ is the $i^{th}$ variant in the pool. For studies with multiple replicates, the final scores are usually obtained by averaging the $s_v$'s across all replicates. Numerous tools have been developed that also implement more sophisticated statistical methods and support more complex experimental designs, such as time-series data, where variant frequencies are tracked over multiple time points to reveal dynamic fitness trends. Below, we introduce the methods in chronological order by publication date. For each, we provide some general background and example studies that have used the tool, a description of supported experimental designs and input data, including sequencing modalities (Fig. 2A–E), and a brief treatment of the statistical method or methods. This information, including links to associated software documentation, is summarised in Table 1.

## Enrich

Enrich (Fowler et al, 2011) was the first standalone DMS scoring tool, having been developed alongside initial deep mutational scanning experimental methods (Fowler et al, 2010). It has been applied in several studies, including protein engineering of A-kinase anchoring proteins (Gold et al, 2013), conformational studies of the HIV-1 Env protein (Heredia et al, 2019), and a study

of MHC-1 antigen loading by the TAPBPR chaperone (McShan et al, 2021). The software is written in Python but is limited to Python 2, which has been officially unsupported since 2020.

Enrich processes FASTQ files to generate counts and attempts to correct sequencing errors by examining agreement between paired-end sequencing reads, if provided. It defines its own format (called "seqid") for representing variants. Enrich only supports amino acid-level variants, and is unable to report nucleotide-level variants.

Enrich is designed exclusively for experiments with two populations, input and selection. Enrich uses the log2-transformed ratio of each variant's selected frequency to its input frequency as the score, called an enrichment ratio. Due to its simplicity and early introduction, this enrichment ratio method remains one of the most widely used approaches to calculate scores in DMS experiments and is often applied by researchers using their own custom analysis scripts, without the use of specialised software.

## dms_tools2

dms_tools2 (Bloom, 2015) is a popular method for virus-based DMS datasets. Based on dms_tools, it estimates site-specific amino acid preferences rather than enrichment scores. The software has been utilised in multiple viral DMS studies, including mutational scans of influenza hemagglutinin (Doud and Bloom, 2016), HIV Envelope (Dingens et al, 2017), and the Zika virus E protein (Kikawa et al, 2023). It is implemented in Python and features additional specialised functions for Jupyter notebook-based data analysis and visualisation (Kluyver et al, 2016).

It enables users to analyse FASTQ files from barcoded-subamplicon sequencing (a high-accuracy, tile-based sequencing protocol) to generate a codon count table, or alternatively, provide their own codon count table to calculate log enrichments, similar to the method used in Enrich, and transforms these enrichments into amino acid preferences for each position.

dms_tools2 computes enrichment scores for two-population experiments, focusing on viral datasets. The primary output is

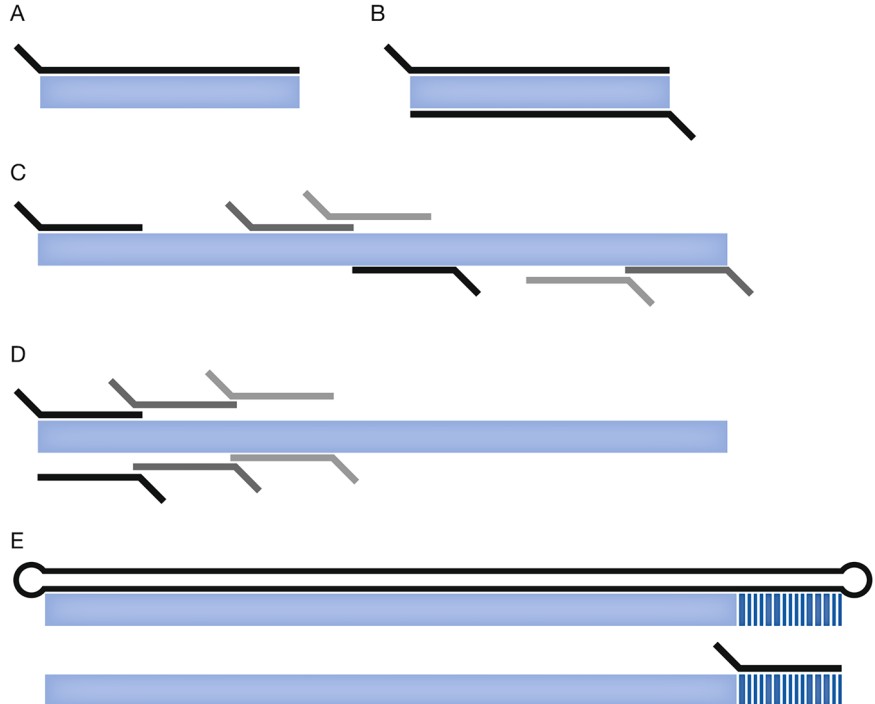

**Figure 2.  Sequencing strategies for DMS experiments.**

The target gene sequence is shown in blue and sequencing reads (or read pairs) are shown in black or grey. Researchers use a variety of different approaches for sequencing DMS populations, including (**A**) single-end direct sequencing, (**B**) overlapping paired-end direct sequencing, (**C**) paired-end shotgun sequencing (e.g., MITE-seq), (**D**) tile-based sequencing, and (**E**) barcode sequencing, which includes a required barcode-variant association long read sequencing reaction (top) followed by short read sequencing for quantification (bottom).

presented as site-specific preferences, which normalises the scores at each position. dms_tools2 has both a standard ratio-based approach as well as a Bayesian method to infer the effects of mutations by modelling the likelihood of observing the count data as a multinomial distribution. When using the Bayesian approach, Dirichlet priors are applied to input frequencies, amino acid preferences, and error rates at each site to incorporate prior knowledge and enable more sophisticated error modelling. The posterior distribution of these parameters is estimated through Markov Chain Monte Carlo sampling, which accounts for potential sequencing and PCR errors.

## Enrich2

Enrich2 (Rubin et al, 2017) is a commonly used DMS scoring tool that analyses diverse experimental datasets, including experiments with two populations and time-series experiments. Enrich2 has been used in diverse studies, including mutational scans of human targets *Src* kinase (Ahler et al, 2019), *MPL* (Bridgford et al, 2020), *BRCA1* (Adamovich et al, 2022), and *PAX6* (McDonnell et al, 2024), as well as non-human targets like the bacterial transporter EfrCD (Meier et al, 2023). It has also been widely used to generate variant counts for downstream analysis, particularly in bin-based assays utilising fluorescence-activated cell sorting (FACS) as a readout (Matreyek et al, 2018), and has been adapted to analyse saturation genome editing data for *CARD11* (Meitlis et al, 2020) and *TP53* (Funk et al, 2025). The software, implemented in Python,

includes a graphical user interface that enables users to input their experimental design and parameters via a point-and-click desktop application. While no longer actively developed, it was recently updated to version 2.0 (Rubin, 2025), providing compatibility with modern versions of Python.

Enrich2 can process FASTQ files generated by single-end direct sequencing, overlapping paired-end direct sequencing, and barcode sequencing, and supports a variety of options for read trimming and filtering. It also supports tabular files containing a matrix of counts as an alternative input method. Enrich2 supports its own HGVS-based (den Dunnen et al, 2016) nomenclature or arbitrary identifiers for describing variants.

Enrich2 offers two classes of methods for calculating variant effect scores: log ratios based on the frequency of each variant for experiments with two populations (usually input and selected) and linear regression for time-series experiments. For time-series experiments, a linear regression model is fitted to the frequencies of each variant across time points, incorporating Poisson variance. Both methods include built-in wild-type normalisation. In experiments with multiple replicates, variant scores are computed independently for each replicate and then combined using a random effects model.

## TileSeqMave

TileSeqMave (Weile et al, 2017) is a computational pipeline for the TileSeq method, designed for tile-based DMS experiments, in

**Table 1. Overview of deep mutational scanning variant scoring tools.**

| Name | Variant populations | | Input sequencing data types | | | | Scoring approach | | Software details | | | Reference |
|---|---|---|---|---|---|---|---|---|---|---|---|---|
| | Two-population | Time-series | Direct | Tiles | Barcodes | Count table | Scoring method | Scoring statistic | Language | Interface[a] | Documentation Link | |
| Enrich | ✓ | | ✓ | | | | Log ratio | Enrichment ratio | Python | CLI | http://depts.washington.edu/sfields/software/enrich/ | Fowler et al, 2011 |
| dms_tools2 | ✓ | | | | | ✓ | Log ratio | Amino acid preference | Python | Module | https://jbloomlab.github.io/dms_tools2/ | Bloom, 2015 |
| Enrich2 | ✓ | ✓ | ✓ | ✓ | ✓ | ✓ | Log ratio, linear regression | Variant score | Python | GUI, CLI | https://enrich2.readthedocs.io/ | Rubin et al, 2017 |
| TileSeqMave | ✓ | ✓ | | ✓ | | ✓ | Log ratio | Enrichment ratio | R | CLI | https://github.com/rothlab/tileseqMave/ | Weile et al, 2017 |
| Fit-Seq2 | ✓ | ✓ | | | | ✓ | Maximum likelihood | Growth fitness | Python | CLI | https://github.com/FangfeiLi05/FitSeq2/ | Li et al, 2023 |
| dms_variants | ✓ | | | | ✓ | ✓ | Log ratio | Functional score | Python | Module | https://jbloomlab.github.io/dms_variants/ | – |
| DiMSum | ✓ | | ✓ | | ✓ | ✓ | Log ratio | Variant fitness | R | CLI | https://github.com/lehner-lab/DiMSum/ | Faure et al, 2020 |
| mutscan | ✓ | | ✓ | | | | Generalised linear model | Fitness score | R | Module | https://fmicompbio.github.io/mutscan/ | Soneson et al, 2023 |
| ACIDES | ✓ | ✓ | | | | ✓ | Maximum likelihood | Variant rank | Python | Module | https://github.com/nemoto-lab/ACIDES/ | Nemoto et al, 2023 |
| Rosace | ✓ | ✓ | | | | ✓ | Log ratio, linear regression | Functional score | R | CLI | https://pimentellab.com/rosace/vignettes/ | Rao et al, 2024 |
| popDMS | ✓ | ✓ | | | | ✓ | Wright-Fisher Model | Selection coefficient | Python | Module | https://github.com/bartonlab/popDMS/ | Hong et al, 2024 |
| gyoza | ✓ | | ✓ | | | | Log ratio | Selection coefficient | Python | CLI | https://github.com/durr1602/gyoza/ | Durand et al, 2025 |

[a]Tools may use a CLI (command line interface), a GUI (graphical user interface), or be imported as a module into a script or analysis environment (e.g., Jupyter notebook or R session).

which libraries are split into defined sub-regions of the target sequence (tiles) to calculate enrichment scores. The pipeline has been utilised in multiple studies, including mutational scans of clinically-relevant genes such as *CBS1* (Sun et al, 2020), *MTHFR* (Weile et al, 2021), and *CHK2* (Gebbia et al, 2024). TileSeqMave is implemented in R and operates through a command-line interface, although potential users should note that elements of the full TileSeqMave pipeline must be installed separately and have their own requirements.

TileSeqMave directly processes count files to compute enrichment scores and offers additional functionalities like a suite of quality control outputs. TileSeqMave outputs data in the MaveDB format (Esposito et al, 2019). TileSeqMave is typically used alongside tileseq_mutcount (https://github.com/RyogaLi/tileseq_mutcount), a Python-based tool that processes FASTQ files from paired-end sequencing of tiles to generate variant counts.

It calculates variant effect scores by determining enrichment ratios between selective and non-selective conditions while applying error regularisation, filtering, and bias correction. The standard error of the calculated enrichment ratio is estimated using two methods. The first approach is bootstrapping, which involves resampling the observed data to measure the variability of the enrichment ratios. The other approach is through a Bayesian regularisation technique, where prior knowledge about the data is introduced to inform the estimation. The score for each variant is derived by normalising and scaling the enrichment ratios.

## Fit-Seq2

Fit-Seq2 (Li et al, 2023) calculates fitness scores for growth-based competitive assays, in two-population or time-series experiments. Fit-Seq2 was preceded by Fit-Seq (Li et al, 2018), which shares the same core functionalities. The program has not been applied to experimental data, but has been tested on simulated data as part of the original publication and includes a simulation framework. The original version of Fit-Seq has analysed non-DMS datasets, such as genetic effects in diploid yeast crosses (Matsui et al, 2022). While the original Fit-Seq was developed in MATLAB, Fit-Seq2 is implemented in Python and has a command-line interface.

Fit-Seq2 operates directly with a count file labelled with arbitrary identifiers. It also requires additional parameters, such as the number of generations and the total effective number of cells in the corresponding sequenced time points.

Fit-Seq2 estimates the fitness of each identifier from read-count time-series data using a likelihood maximisation approach, ensuring that the expected lineage trajectories align with the observed data. It models the expected number of cells for each variant based on the cell count at previous time points, the growth rate, a global correction factor for the mean fitness of all variants, and noise factors that account for experimental conditions. Fit-Seq2 defines a likelihood function as the joint probability of observing the sequence of counts, given the initial cell number and fitness. It then iteratively estimates the fitness of each variant and the initial cell count to maximise this likelihood function.

## dms_variants

dms_variants (https://github.com/jbloomlab/dms_variants) is a tool for analysing barcoded DMS datasets developed by the same group that wrote dms_tools and dms_tools2 (Bloom, 2015). Like its predecessors, the software has been used for viral studies, including a mutational scan of the receptor binding domain of the SARS-CoV-2 spike protein (Starr et al, 2020), and it also has robust support for simulating datasets. It is designed to run within Python-based Jupyter notebooks, and many example analyses are provided.

dms_variants processes FASTQ files containing barcode reads and converts them into variants with a user-provided codon-variant table, which links each substitution to its corresponding barcode. It also supports in-frame (codon-length) deletions.

Like dms_tools2, dms_variants focuses on studies that have two time points. The scores are calculated using a log ratio of ratios, but instead of using a variant frequency, the count of each variant is divided by the total wild-type count in that time point. dms_variants also has the ability to fit a global epistasis model (Otwinowski et al, 2018) and model experimental noise.

## DiMSum

DiMSum (Faure et al, 2020) is another popular tool for DMS scoring that was the first to apply empirical Bayes methods similar to those that underpin RNA-seq analysis (Robinson et al, 2010; Anders and Huber, 2010) to functional assays. The method has been used in many studies, including an assay for amyloid beta aggregation (Arutyunyan et al, 2025) and mutational scans of KRAS (Weng et al, 2024) and the *Trypanosoma brucei* gene KREPB4 (McDermott et al, 2024). It is implemented in R and has a command-line interface.

DiMSum processes FASTQ files from single-end sequencing, paired-end sequencing, and barcode sequencing into count files. It also performs quality control and read trimming steps if desired. The read counting and scoring steps of the pipeline are logically separated into two components ("WRAP" and "STEAM"), and can be run independently or in sequence.

DiMSum is designed exclusively for experiments with two populations, an input before selection and an output following selection. Variant effect scores are calculated as log ratios based on count frequencies. DiMSum assumes that counts follow the Poisson distribution, where the variance of the sequencing counts is equal to the mean. To address the issue of overdispersion, DiMSum expands error terms to account for error sources such as sequencing errors and experimental setup differences and estimates these terms via bootstrapping. It integrates scores across replicates through a weighted average.

## mutscan

mutscan (Soneson et al, 2023) is a DMS scoring tool designed to improve data processing efficiency, offering faster computation, lower memory usage, and effective multi-core utilisation. The program has been used in a study assaying protein-protein interactions (Bendel et al, 2024), but only for FASTQ file processing. It is implemented in R and can be run via script or an interactive R session.

mutscan processes FASTQ files, supporting both single-end and paired-end sequencing. It includes quality assessment features and allows users to customise various parameters for FASTQ processing.

mutscan is designed to handle experiments with two populations and calculates fitness scores by normalising counts and fitting them to factors such as replicates or time points using generalised linear models. The edgeR option estimates dispersion parameters and fits a negative binomial model to calculate the scores, while the limma option transforms the counts with limma-voom to adjust the mean-variance relationship and weight each observation. limma then applies linear models to the transformed data to estimate the scores and uses empirical Bayes statistics to moderate the standard errors of these estimates. limma and edgeR are widely used for differential expression analysis in RNA-seq experiments, but there are fundamental differences between DMS and RNA-seq as highlighted in the literature (Rao et al, 2024). One key assumption under differential expression analysis is that the majority of genes are not differentially expressed between conditions. This assumption is violated in DMS studies, where a substantial portion of the mutations are expected to significantly alter protein function.

## ACIDES

ACIDES (Accurate Confidence Intervals for Directed Evolution Scores) (Nemoto et al, 2023) is a DMS scoring tool that leverages statistical inference and simulations to improve variant effect score estimations. To date, it has not been applied to experimental datasets, but has been tested on simulated data as part of the original publication. The software is written in Python and should be imported into the user's scripts or notebooks as a module.

ACIDES consumes a count file labelled with arbitrary identifiers, so it requires an additional tool to process FASTQ files.

ACIDES models the variant counts using a negative binomial distribution. It calculates the log-likelihood for each variant across different time points, and a selectivity score is inferred using maximum likelihood estimation. Apart from the estimated scores, ACIDES introduces the corrected ranking of variants to improve the reliability of variant ranking. The naive ranking is obtained by sorting the estimated scores of each variant, and is corrected by using in silico simulations. This focus on variant ranking rather than scores is unique amongst the methods reviewed here.

## Rosace

Rosace (Rao et al, 2024) is a DMS scoring tool designed for analysing competitive growth-based assays, including two-population and time-series experiments, introducing a novel approach to incorporate positional information into variant effect scores. The package has been used in several studies, including mutational scans of G protein-coupled receptors (Howard et al, 2024) and MET receptor tyrosine kinase (Estevam et al, 2024). Rosace also includes a companion simulation framework, Rosette. The software is implemented in R and features a command-line interface. It can be installed directly or run using the provided Docker container image.

Rosace processes variant count files labelled in a specific HGVS-based variant nomenclature. For analysis from FASTQ files, the developers created the Dumpling pipeline using Snakemake (Mölder et al, 2021) that can process directly sequenced libraries.

Rosace uses a time-dependent linear function to model the log ratios of normalised counts for each variant. It assumes that variants at the same position are likely to exhibit similar functional effects and introduces a position-specific term to account for these shared effects. The mean and variance of each variant are estimated within a hierarchical Bayesian framework. To enhance the stability of variance estimates, Rosace groups variants based on their mean counts across different time points and replicates, and further includes a group-specific variance parameter in the model. This approach aims to improve the statistical power of detecting phenotypic consequences and control the false discovery rate through shrinkage, which shares information across parameters.

## popDMS

popDMS (Hong et al, 2024) is a DMS scoring tool developed to analyse two-population experiments using the Wright-Fisher model, an evolutionary model from population genetics. The original publication applied the method to a diverse collection of 28 DMS datasets from the literature. It is implemented in Python and C++ and is executable in Python from the user's own scripts or analysis notebooks.

popDMS accepts input in the form of codon counts formatted for dms_tools2 or sequence counts formatted for MaveDB.

popDMS uses the Wright-Fisher model to infer the effects of individual mutations. It utilises a Bayesian framework to estimate the selection coefficient for each variant. Beyond assessing individual mutation effects, popDMS also enables the estimation of pairwise epistatic interactions between variants at different sites. It introduces a Gaussian prior distribution for every score that acts as a regularisation mechanism to prevent model overfitting, particularly for experiments involving a large number of parameters. The score of each variant is estimated by maximising the posterior distribution, which effectively joins the information across multiple experimental replicates.

### gyōza

gyōza (Durand et al, 2025) is a modular scoring pipeline implemented using the Snakemake framework that includes support for analysing multiple experiments or scans of multiple loci in a single run. The tool was tested using the same small test dataset used to demonstrate DiMSum (Faure et al, 2020), and the authors report on the performance of internal tests using large unpublished datasets. As a Snakemake pipeline, gyōza wraps several other tools for read processing and quality control reporting. The DMS-specific and other custom parts of the pipeline are implemented in Python using Jupyter notebooks that are executed when the pipeline is run.

gyōza analyses overlapping paired-end reads in FASTQ format. It also supports tile-based designs by specifying each tile as a different target. Variants are called by enumerating all possible single amino acid variants and comparing these sequences to the merged input reads, so libraries with multiple amino acid changes are not supported by default.

The software uses a basic log2 fold-change based method to calculate variant functional scores. gyōza has the optional ability to normalise scores based on the number of cellular generations for experimental designs where this information is known (e.g., yeast growth assays), which allows the score to be interpreted as a selection coefficient. Scores are further normalised by subtracting the median synonymous variant score.

# Comparison of variant scoring tools

Comparing DMS scoring tools reveals a range of strengths and limitations shaped by their diverse needs. Categorising variant scoring tools is challenging because they aim to achieve a similar outcome, but the heterogeneity in DMS experimental designs necessitates the use of different approaches. Methods differ substantially in their statistical models, input data requirements, and underlying implementations, making each better suited for specific analyses. The computational resources required to execute each tool on a given dataset also vary due to implementation differences. DMS scoring tools use data frames to store and manipulate datasets, meaning that memory requirements scale linearly with the number of variants. For most workflows, runtime is dominated by FASTQ file processing (i.e., the process is I/O bound), since the files are large and the calculations performed to score each variant are not particularly intensive.

## Supported experimental designs

Pooled assays used for DMS can measure various readouts, including cell growth, survival, protein binding, enzymatic activity, and fluorescence. However, designs generally follow three main formats: two-population experiments with an input and a single post-selection output, time-series experiments with an input and multiple outputs over time, and bin-based experiments with multiple discrete output bins sorted using FACS. Therefore, choosing a tool tailored to the specific experimental readout is crucial for proper data analysis.

The most common DMS design is the two-population experiment, and it has the broadest support from scoring tools. DiMSum and Enrich2 are both routinely used except for viral experiments, where dms_tools2 and dms_variants are more commonly employed. For time-series experiments, Enrich2, Fit-Seq2, ACIDES, and Rosace can be applied, though Enrich2 is the oldest and remains the most popular choice among researchers for now. Currently, no dedicated tool exists for analysing readouts from FACS/bin-based experiments, but experiment-specific code has been made available as part of relevant studies (Matreyek et al, 2018; Aslanzadeh et al, 2024).

## Input and output file formats

DMS scoring tools typically accept two types of inputs: FASTQ files from high-throughput sequencing, which provide either direct sequences of variants or associated barcodes, or tabular count files derived from those FASTQ files that provide integer counts per variant or sequence. Most tools, including Enrich, dms_tools2, Enrich2, TileSeqMave, DiMSum, and mutscan, support both input formats, while Fit-Seq2, ACIDES, Rosace, and popDMS are limited to processing count files.

All DMS scoring tools generate numeric scores representing variant effects, though the nature of these scores may differ and, consequently, authors refer to these scores using different terms, such as enrichment ratio, variant score, growth fitness, functional score, selectivity score, variant rank, or selection coefficient. Tools provide a variety of additional values such as variance estimates or $P$ values that are specific to the statistical method and implementation. The output data files themselves are usually comma- or tab-separated tabular files that can be easily opened using a text editor, spreadsheet program, or data science programming environment. Notably, DiMSum and TileSeqMave are unique in providing an option to generate output files that are ready to be uploaded to the community database MaveDB (Rubin et al, 2025).

## Data visualisation

In addition to providing estimated scores for each variant, many tools offer a diverse range of visualisation options to help users interpret the results and assess the quality of their data. The most common type of visualisation specific to DMS is the "sequence-function map" or "variant effect map", which is a type of heatmap where each cell is a variant and the rows and columns correspond to variant amino acids and positions in the protein sequence.

Enrich, Enrich2, Rosace, and popDMS offer built-in heatmaps to visualise variant scores across positions (Fig. 3A–D), with Rosace providing additional violin and density plots to display score distributions. Enrich2 also implements a variety of diagnostic plots, such as variant count histograms, per-library diversity heatmaps, and representative linear fits for time-series experiments.

DiMSum produces an HTML report containing a summary and multiple diagnostic plots, including hexagonal heatmaps that link estimated scores with input variant counts.

mutscan visualises statistical test results for enrichment between two conditions compared to wild-type sequences, using mean-difference and volcano plots.

dms_tools2 and dms_variants specialise in site-specific preferences and can integrate protein structural information into their visualisations (Fig. 3E,F). The documentation for dms_variants includes examples of a variety of diagnostic plots as well. Researchers who want to generate site-specific preference plots from other tools can use the standalone dms_logo (https://jbloomlab.github.io/dmslogo/) package used by dms_variant, or an alternative such as Logomaker (Tareen and Kinney, 2020).

ACIDES provides a visualisation of the corrected rankings for the variants (Fig. 3G), although it does not produce more traditional DMS plots like heatmaps itself. Tools like Fit-Seq2 and TileSeqMave lack built-in visualisation capabilities, requiring supplementary tools for visualising results.

Although it is not an analysis tool, the MaveDB database provides visualisation options for uploaded data, displaying scores as heatmaps or density plots (Fig. 3H), facilitating data interpretation and exploration. Researchers seeking to explore their data on a protein structure can use tools such as dms-view (Fig. 3I) (Hilton et al, 2020) or dms-viz (Hannon and Bloom, 2024). Additionally, well-established structural biology software such as UCSF ChimeraX (Meng et al, 2023) can be used for this purpose.

## Statistical assumptions

Enrich, dms_tools2, dms_variants, DiMSum, mutscan, and gyōza are designed for studies with two time points, whereas the remaining tools (Enrich2, TileSeqMave, Fit-seq2, ACIDES, Rosace, and popDMS) can be used for studies with multiple time points.

Each tool has unique estimations for variant scores and corresponding error terms. Tools like dms_tools2, Rosace, TileSeqMave, and popDMS incorporate Bayesian approaches for the estimation process that borrow prior information. Other tools

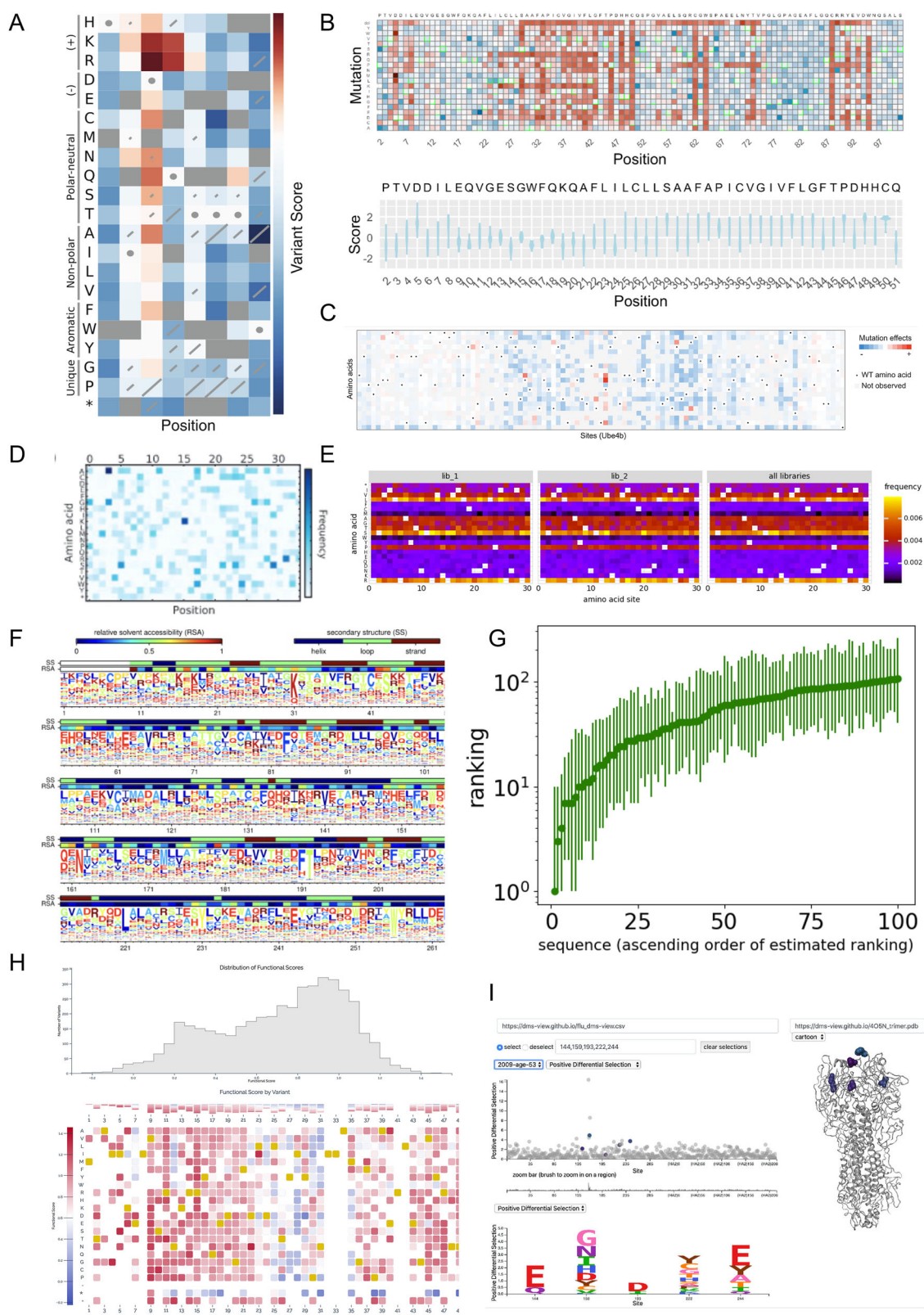

**Figure 3.   Score visualisations.**

**(A)** Sequence-function map showing scores and standard errors for the YAP1 WW domain (Fowler et al, 2010) obtained from Enrich2. **(B)** Variant effect heatmap and score density plots for OCT1 (Yee et al, 2024) from Rosace. **(C)** Variant effect heatmap for murine UBE4B (Starita et al, 2013) from popDMS. **(D)** A heatmap of the frequency of each position–mutation combination from Enrich's example dataset. **(E)** Heatmap of mutation frequencies for simulated codon variants from dms_variants. **(F)** Visualisation of site-specific preferences for a Tn5 transposon (Melnikov et al, 2014) from dms_tools. **(G)** Corrected variant ranking for the yeast two-hybrid assay targeting BRCA1 (Starita et al, 2015) presented by ACIDES. **(H)** Score histogram and variant effect heatmap for PTEN VAMP-seq (Matreyek et al, 2018) from MaveDB. **(I)** Interactive visualisation for exploring DMS scores on the protein structure of influenza hemagglutinin (Lee et al, 2019) from dms-view.

mainly use frequentist frameworks for inference. ACIDES and Fit-Seq2 both use maximum likelihood for estimating scores. For experiments with multiple replicates, dms_tools2 combines the replicates by taking the average, whereas DiMSum combines replicates using the estimated error term and the weighted mean approach. Enrich2 and Rosace integrate replicates through more complicated statistical modelling.

## Software implementation and documentation

Most DMS scoring tools are built in Python or R and need to be installed from a source code repository following the instructions provided. A typical setup involves creating a virtual environment using Anaconda or a similar tool, installing the software along with its dependencies within that environment, and running the tool from there. This approach helps manage package versions, prevent conflicts with other installed software, and ensure reproducibility across different systems. Methods are usually tested on multiple datasets, either experimental or simulated, and are generally well-documented, providing clear guidance for users to effectively run them on their own data. Many of the packages include example analyses in interactive data analysis notebooks that can be adapted by end users. However, as with all software, DMS scoring tools may become more difficult to run if they are not continuously maintained by the authors, and this is a particularly acute problem for academic software written by graduate students or postdocs (Prlić and Procter, 2012).

# Other tools used for data processing

Additional software has been written for working with DMS data, which can be broadly categorised into standalone variant counting tools for FASTQ processing, tools for associating variants to barcodes based on long read sequencing data, and tools for further processing DMS scores to help researchers derive insights from their data.

## Standalone variant counting tools

Some tools cannot process FASTQ files directly and require other software to process and generate count files from sequence data. While many of the above methods can generate counts alone, several tools have been developed for this specific purpose and do not provide a scoring function. These include satmut_utils (Hoskins et al, 2023), which uses a more sophisticated variant calling and error correction strategy than some all-in-one tools, and AnalyzeSaturationMutagenesis (Yang et al, 2023), which is the updated version of ORFCall, designed to count DMS variants from

shotgun sequencing data produced by methods like MITE-seq (Melnikov et al, 2014). AnalyzeSaturationMutagenesis is distributed as part of the Genome Analysis Toolkit (GATK) (McKenna et al, 2010).

## Barcoded library tools

A required step in barcode-based DMS workflows is to link the barcodes to their corresponding variant. This is generally done using long read sequencing to generate single reads containing both the barcode and full-length variant sequence, followed by analysis to produce a barcode-variant map file that is required by the DMS scoring software. Several tools are available for performing this operation, including alignparse (Crawford and Bloom, 2019), PacRAT (Yeh et al, 2022), and Pacybara (Weile et al, 2024).

## Post-processing and modelling tools

Various factors during library preparation, screening, and sequencing steps in DMS experiments lead to incomplete variant effect maps. The Human Protein Variant Effect Map Imputation Toolkit (Wu et al, 2019) addresses this problem by imputing missing values based on predictive features, building on previous work (Weile et al, 2017). It offers a user-friendly web application, making it accessible to users with limited programming expertise (http://impute.varianteffect.org/). FUSE (Functional Substitution Estimation) (Yu et al, 2024) is another approach for imputing missing data. It is trained on secondary structure information and a broad set of DMS datasets. Like the previous method, FUSE provides a web interface for ease of use (https://tyu7.shinyapps.io/FUSE/).

Several tools have been developed to derive more biologically-relevant measurements from DMS scores, including MAVE-NN (Tareen et al, 2022), LANTERN (Tonner et al, 2022), and MoCHI (Faure and Lehner, 2024). These tools process variant effect scores, extract relevant features, and estimate biological or biochemical properties from experimental data. They can play an important role in interpreting DMS data by contributing to the understanding of the variant effects being measured.

For researchers interested in evolutionary studies, phydms (Hilton et al, 2017) analyses DMS data in the context of phylogenetic trees, using codon models from molecular phylogenetics to investigate whether scores are concordant with evolutionary data.

# Challenges and future directions

While significant progress has been made in developing variant scoring tools and analysis pipelines for DMS experiments, several

challenges remain in the field. Researchers often use the scoring tool they are most familiar with, yet each tool offers unique features that may better suit specific experimental datasets, and more recent tools may have superior statistical approaches. The lack of standardisation in output formats and configuration files exacerbates this problem, creating a significant cost for users who want to explore alternative options for variant scoring. Furthermore, this creates challenges for comparisons and benchmarking. A robust and fair comparison between the various statistical methods presented here remains an outstanding need for the field, which is made more daunting by the non-uniform requirements for high-throughput sequencing strategies and experimental designs.

Variant effect scores are frequently reported in study-specific formats, making downstream interpretation difficult, although the community database MaveDB provides a de facto standard data format for researchers and tool developers to target. Additionally, there is no consensus on how to estimate variability for scores, and variance estimates or confidence intervals are not computed by all pipelines, substantially complicating both comparisons of results reported in the literature and clinical applications of DMS (Starita et al, 2017; Fowler et al, 2023). Improving and standardising the representation of variant scores will help further empower the use of DMS scores for machine learning and AI-based approaches, including clinical variant effect prediction (Notin et al, 2023; Livesey and Marsh, 2023).

There are also clear opportunities for the field. Many important experimental types, such as bin-based assays or saturation genome editing, lack robust software support and are primarily analysed using one-off scripts that can be difficult to reuse or rerun. In addition, there is a need for ancillary tools to improve the utility of DMS scores, such as those that can perform imputation or quality assessment.

The widespread application of DMS scores for clinical variant classification, public health genomics, protein engineering, and other diverse applications is encouraging. We look forward to the next generation of computational tools for DMS that will better integrate statistical rigour, ease of use, and support for diverse experimental designs, enabling DMS to fully deliver on its promise for biology and medicine.

## Peer review information

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

## Acknowledgements

AFR received funding from National Human Genome Research Institute grants UM1HG011969 and RM1HG010461. JAM is supported by the European Research Council (grant agreement No. 101001169) and by the Medical Research Council (MRC) Human Genetics Unit core grant (MC_UU_00035/9). BP is supported by National Health and Medical Research Council Investigator grant GNT2025641. XJ is supported by a CSL Translational Data Science Scholarship, University of Melbourne Research Scholarship, and WEHI Scientific Excellence Scholarship. This project received grant funding from the Australian government.

## Author contributions

**Hasan Çubuk**: Conceptualization; Data curation; Writing—original draft; Writing—review and editing. **Xinyi Jin**: Data curation; Funding acquisition; Visualisation; Writing—original draft; Writing—review and editing. **Belinda Phipson**: Supervision; Funding acquisition; Writing—original draft; Project administration; Writing—review and editing. **Joseph A Marsh**: Conceptualization; Supervision; Funding acquisition; Writing—original draft; Project administration; Writing—review and editing. **Alan F Rubin**: Conceptualization; Data curation; Supervision; Funding acquisition; Visualisation; Writing—original draft; Project administration; Writing—review and editing.

## Disclosure and competing interests statement

The authors declare no competing interests.

