## [Peer Review File · Molecular Systems Biology]

Variant Scoring Tools for Deep Mutational Scanning

Hasan Çubuk, Xinyi Jin, Belinda Phipson, Joseph Marsh, and Alan Rubin

Corresponding author(s): Alan Rubin (alan.rubin@wehi.edu.au) , Joseph Marsh (joseph.marsh@ed.ac.uk)

Review Timeline:

Submission Date:	8th Apr 25
Editorial Decision:	6th Jun 25
Revision Received:	18th Jun 25
Accepted:	22nd Jul 25

Editor: Poonam Bheda

Transaction Report:

6th Jun 2025

Manuscript Number: MSB-2025-13005

Title: Variant Scoring Tools for Deep Mutational Scanning

Dear Dr Rubin,

Thank you for your submission to Molecular Systems Biology. Please find below the two sets of comments I have now received regarding your piece. As you will see, the referees are all positive about its timeliness and suitability for publication. However, they do offer some suggestions on how to improve the review for your readers as follows:

- include some perspective on how deep learning could be used for predicting DMS results, especially in such cases of rapid viral evolution (Reviewer 1)
- report computational requirements such as memory, typical run times, etc (Reviewer 2)
- adding workflow information for each approach as e.g. an addition to your Table 1 (Reviewer 2)

If you would like to discuss any of these points, I am available via email or we can schedule a call.

- 1) a .doc formatted version of the manuscript text (including Figure legends and tables, no track changes)
- 2) Separate figure files. Please remove all figures from main manuscript file and leave only main figure legends placed after the references.

We will send the article to our graphic artist who will edit/re-draw Figures 1 and 2 for style. Therefore, please ensure the information shown is scientifically accurate and upload the file as a PDF (or SVG, or EPS), PowerPoint or Keynote in which the labels and objects are still editable. For figures created using Adobe Illustrator, please send the Illustrator (.ai) file. You can also send these to me by email (or share via link for files that are too big) so that we can already send these to the graphic designer to prevent delay in publishing your manuscript.

- 3) Callouts in the manuscript for the panels for Fig. 2 (A-E) and 3 (A-I).
- 4) Up to 5 keywords
- 5) All funding sources. Currently grant funding from the Australian government is missing in our manuscript submission system.
- 6) As part of the EMBO Publications transparent editorial process initiative (see our policy here: https://www.embopress.org/transparent-process#Review_Process), Molecular Systems Biology will publish online a Peer Review File (PRF) to accompany accepted manuscripts. This file will be published in conjunction with your paper and will include the anonymous referee reports, your point-by-point response and all pertinent correspondence relating to the manuscript. Let us know whether you agree with the publication of the PRF and as here, if you want to remove or not any figures from it prior to publication.
- 7) Please provide a point-by-point letter INCLUDING my comments as well as the reviewer's reports and your detailed responses (as Word file).

If you have any questions, please don't hesitate to ask. I look forward to seeing the revised manuscript.

Yours sincerely,

Poonam Bheda, PhD
Scientific Editor
Molecular Systems Biology

Reviewer #1:

This manuscript evaluates the strengths and weaknesses of variant scoring tools used in deep mutational scanning (DMS). It addresses current challenges such as the need for standardized analysis protocols and sustainable software maintenance, while also highlighting opportunities for future method development. The goal of this review is to promote the application and adoption of DMS, thereby enhancing biological understanding and improving clinical translation. The review is well-written. However, there is a lack of discussion in the challenges of experimental DMS to rapidly respond to fast viral evolution and the potential future directions in using experimental DMS and its variant scores to develop new in-silico DMS. This perspective should be included to enhance the quality of this review.

The `dms_tools2` (Bloom, 2015) played a significant role during the COVID-19 pandemic but the experimental DMS are still a bottleneck in responding to fast evolving viruses. In May 2020, a topological deep learning (TDL) approach predicted in-silico DMS of SARS-CoV-2 spike RBD was made available [1], which is four months earlier than the first experimental SARS-CoV-2 DMS made public. This work predicted the important mutation sites 452 and 501, which were later found to host the key mutations of all prevailing variants, alpha, beta, gamma, delta, theta, mu, and omicron, BA.2, BA.4, and BA.5, etc. TDL-based DMS successfully predicted the emerging dominance of Omicron BA.2 in early February of 2022 [4], that was confirmed by the WHO in late March of 2022.

Nonetheless, experimental DMS and its variant scores, such as log enrichment ratios, remain crucial to training data for enhancing TDL approaches in predicting in-silico DMS of the SARS-CoV-2 spike RBD. Seven experimental SARS-CoV-2 DMS datasets were utilized in [3], which further reinforced the accuracy and robustness of TDL-DMS models in forecasting SARS-CoV-2 variants and designing effective antibodies and vaccines. Despite the availability of experimental DMS data, TDL approaches still depend on high-quality 3D PDB structures to leverage experimental DMS for predicting in-silico DMS. Recently, AlphaFold 3 has shown the capability to mitigate this challenge when 3D PDB structures are unavailable [4]. Utilizing AlphaFold 3's SARS-CoV-2 structures has expanded TDL approaches to predict the impacts of viral mutations, monitor virus evolution, anticipate emerging dominant variants, and guide the development of new vaccines. The ongoing advancements in experimental structure determination methods, such as significant improvements in cryo-electron microscopy and tomography, remain essential to provide new 3D structures in order to enhance generalization capability of in-silico DMS methods in future.

[1] J. Chen, R. Wang, M. Wang, and G.-W. Wei, "Mutations strengthened SARS-CoV-2 infectivity," *Journal of molecular biology*, vol. 432, no. 19, pp. 5212-5226, 2020.

[2] Chen, J., & Wei, G. W. (2022). Omicron BA. 2 (B. 1.1. 529.2): high potential for becoming the next dominant variant. *The journal of physical chemistry letters*, 13(17), 3840-3849.

[3] Chen, J., Woldring, D. R., Huang, F., Huang, X., & Wei, G. W. (2023). Topological deep learning based deep mutational scanning. *Computers in biology and medicine*, 164, 107258.

[4] JunJie Wee, Guo-Wei Wei, Rapid response to fast viral evolution using AlphaFold 3-assisted topological deep learning, *Virus Evolution*, 2025; veaf026, <https://doi.org/10.1093/ve/veaf026> .

Reviewer #2:

In this work, Çubuk, Jin, and collaborators review a broad array of computational methods for scoring variant effects in deep mutational scanning experiments. Their review covers the statistical approaches of each method, supported data types and experimental designs, how they are implemented computationally, some key assumptions, and associated methods for data visualization. They also highlight several opportunities for improvement in the field, such as a lack of methods designed for particular experimental approaches and the need for high-quality statistical comparisons of different methods.

Overall, this paper provides a comprehensive overview of methods for analyzing DMS data. I am sure that this work will prove useful both for researchers looking to analyze DMS data and those who focus on methods development. The breadth of the authors' analysis is especially notable, spanning from the very earliest approaches (Enrich, `dms_tools`) to ones developed just within the past year. The section on post-processing and modeling tools is a particularly nice addition that directs attention toward methods that can help to extract even more information from rich DMS data sets. It's great that the authors have broadly covered the ecosystem of computational methods for DMS data.

Below, I have a few suggestions that may strengthen the paper.

Main comments

1. As the authors have noted, benchmarking different methods for variant effect estimation is challenging for multiple reasons. While statistical comparisons are quite difficult, it may be helpful for users to give a sense of the computational resources required to run the software. For example, knowledge of typical memory requirements, run times on example data sets, or even computational complexity could be helpful to understand how these different approaches scale. Of course, as the authors have noted, one must also keep in mind that the data types and analyses performed by each method are different. If reporting computational metrics across methods is too challenging for the present work, then perhaps the authors could comment in general on issues related to computational efficiency. Scalability could become important as experiments also grow in scope.

2. It may be helpful to give a sense of the workflow for each approach, especially for readers who are not well-versed in computational analyses. This has been described already in the text for most methods, but it could be helpful to make these descriptions more systematic, for example by incorporating how the software is run in Table 1 (e.g., GUI, notebook, command line interface).

6th Jun 2025

Manuscript Number: MSB-2025-13005

Title: Variant Scoring Tools for Deep Mutational Scanning

Dear Dr Rubin,

Thank you for your submission to Molecular Systems Biology. Please find below the two sets of comments I have now received regarding your piece. As you will see, the referees are all positive about its timeliness and suitability for publication. However, they do offer some suggestions on how to improve the review for your readers as follows:

- include some perspective on how deep learning could be used for predicting DMS results, especially in such cases of rapid viral evolution (Reviewer 1)
- report computational requirements such as memory, typical run times, etc (Reviewer 2)
- adding workflow information for each approach as e.g. an addition to your Table 1 (Reviewer 2)

If you would like to discuss any of these points, I am available via email or we can schedule a call.

Thank you for this helpful summary. We have prepared our point by point response with the author responses in blue.

1) a .doc formatted version of the manuscript text (including Figure legends and tables, no track changes)

Done

2) Separate figure files. Please remove all figures from main manuscript file and leave only main figure legends placed after the references.

Done

We will send the article to our graphic artist who will edit/re-draw Figures 1 and 2 for style. Therefore, please ensure the information shown is scientifically accurate and upload the file as a PDF (or SVG, or EPS), PowerPoint or Keynote in which the labels and objects are still editable. For figures created using Adobe Illustrator, please send the Illustrator (.ai) file. You can also send these to me by email (or share via link for files that are too big) so that we can already send these to the graphic designer to prevent delay in publishing your manuscript.

3) Callouts in the manuscript for the panels for Fig. 2 (A-E) and 3 (A-I).

Done

4) Up to 5 keywords

We have selected the following keywords:

deep mutational scanning, multiplexed assays of variant effect, functional genomics, bioinformatics, software

5) All funding sources. Currently grant funding from the Australian government is missing in our manuscript submission system.

The Australian government does not publish grant numbers for this funding scheme and instead requires the generic funding acknowledgement listed. We have added the funding from the Australian Medical Research Future Fund (MRFF) to the submission system.

6) As part of the EMBO Publications transparent editorial process initiative (see our policy here: https://www.embopress.org/transparent-process#Review_Process), Molecular Systems Biology will publish online a Peer Review File (PRF) to accompany accepted manuscripts. This file will be published in conjunction with your paper and will include the anonymous referee reports, your point-by-point response and all pertinent correspondence relating to the manuscript. Let us know whether you agree with the publication of the PRF and as here, if you want to remove or not any figures from it prior to publication.

Yes, we agree to the publication of the PRF.

7) Please provide a point-by-point letter INCLUDING my comments as well as the reviewer's reports and your detailed responses (as Word file).

Done

If you have any questions, please don't hesitate to ask. I look forward to seeing the revised manuscript.

Yours sincerely,

Poonam Bheda, PhD

Scientific Editor

Molecular Systems Biology

Reviewer #1:

This manuscript evaluates the strengths and weaknesses of variant scoring tools used in deep mutational scanning (DMS). It addresses current challenges such as the need for standardized analysis protocols and sustainable software maintenance, while also highlighting opportunities for future method development. The goal of this review is to promote the application and adoption of DMS, thereby enhancing biological understanding and improving clinical translation. The review is well-written. However, there is a lack of discussion in the challenges of experimental DMS to rapidly respond to fast viral evolution and the potential future directions in using experimental DMS and its variant scores to develop new in-silico DMS. This perspective should be included to enhance the quality of this review.

The dms_tools2 (Bloom, 2015) played a significant role during the COVID-19 pandemic but the experimental DMS are still a bottleneck in responding to fast evolving viruses. In May 2020, a topological deep learning (TDL) approach predicted in-silico DMS of SARS-CoV-2 spike RBD was made available [1], which is four months earlier than the first experimental SARS-CoV-2 DMS made public. This work predicted the important mutation sites 452 and 501, which were later found to host the key mutations of all prevailing variants, alpha, beta, gamma, delta, theta, mu, and omicron, BA.2, BA.4, and BA.5, etc.

TDL-based DMS successfully predicted the emerging dominance of Omicron BA.2 in early February of 2022 [4], that was confirmed by the WHO in late March of 2022.

Nonetheless, experimental DMS and its variant scores, such as log enrichment ratios, remain crucial to training data for enhancing TDL approaches in predicting in-silico DMS of the SARS-CoV-2 spike RBD. Seven experimental SARS-CoV-2 DMS datasets were utilized in [3], which further reinforced the accuracy and robustness of TDL-DMS models in forecasting SARS-CoV-2 variants and designing effective antibodies and vaccines. Despite the availability of experimental DMS data, TDL approaches still depend on high-quality 3D PDB structures to leverage experimental DMS for predicting in-silico DMS. Recently, AlphaFold 3 has shown the capability to mitigate this challenge when 3D PDB structures are unavailable [4]. Utilizing AlphaFold 3's SARS-CoV-2 structures has expanded TDL approaches to predict the impacts of viral mutations, monitor virus evolution, anticipate emerging dominant variants, and guide the development of new vaccines. The ongoing advancements in experimental structure determination methods, such as significant improvements in cryo-electron microscopy and tomography, remain essential to provide new 3D structures in order to enhance generalization capability of in-silico DMS methods in future.

[1] J. Chen, R. Wang, M. Wang, and G.-W. Wei, "Mutations strengthened SARS-CoV-2 infectivity," *Journal of molecular biology*, vol. 432, no. 19, pp. 5212-5226, 2020.

[2] Chen, J., & Wei, G. W. (2022). Omicron BA. 2 (B. 1.1. 529.2): high potential for becoming the next dominant variant. *The journal of physical chemistry letters*, 13(17), 3840-3849.

[3] Chen, J., Woldring, D. R., Huang, F., Huang, X., & Wei, G. W. (2023). Topological deep learning based deep mutational scanning. *Computers in biology and medicine*, 164, 107258.

[4] JunJie Wee, Guo-Wei Wei, Rapid response to fast viral evolution using AlphaFold 3-assisted topological deep learning, *Virus Evolution*, 2025; veaf026, <https://doi.org/10.1093/ve/veaf026>.

We thank the reviewer for their feedback and are glad they found the manuscript to be well-written. We agree that *in silico* DMS is important but feel that it is out of scope for our paper that focuses entirely on tools for experimental DMS. Variant effect prediction (including methods for viral sequences) is a large and fast-moving field deserving its own dedicated review.

Reviewer #2:

In this work, Çubuk, Jin, and collaborators review a broad array of computational methods for scoring variant effects in deep mutational scanning experiments. Their review covers the statistical approaches of each method, supported data types and experimental designs, how they are implemented computationally, some key assumptions, and associated methods for data visualization. They also highlight several opportunities for improvement in the field, such as a lack of methods designed for particular experimental approaches and the need for high-quality statistical comparisons of different methods.

Overall, this paper provides a comprehensive overview of methods for analyzing DMS data. I am sure that this work will prove useful both for researchers looking to analyze DMS data and those who focus on methods development. The breadth of the authors' analysis is especially notable, spanning from the very earliest approaches (Enrich, *dms_tools*) to ones developed just within the past year. The section on post-processing and modeling tools is a particularly nice addition that directs attention toward methods that can help to extract even more information from rich DMS data sets. It's great that the authors have broadly covered the ecosystem of computational methods for DMS data.

We thank the reviewer for this positive assessment and are glad that the thoroughness and extra content on post-processing and modelling was appreciated.

Below, I have a few suggestions that may strengthen the paper.

Main comments

1. As the authors have noted, benchmarking different methods for variant effect estimation is challenging for multiple reasons. While statistical comparisons are quite difficult, it may be helpful for users to give a sense of the computational resources required to run the software. For example, knowledge of typical memory requirements, run times on example data sets, or even computational complexity could be helpful to understand how these different approaches scale. Of course, as the authors have noted, one must also keep in mind that the data types and analyses performed by each method are different. If reporting computational metrics across methods is too challenging for the present work, then perhaps the authors could comment in general on issues related to computational efficiency. Scalability could become important as experiments also grow in scope.

We agree that understanding the computational resource requirements is important for readers. However, as the reviewer points out, given the diversity of methods (especially the fact that many have mutually-exclusive input data requirements), it's not feasible to generate and compare metrics. Instead, as suggested we have added the following text on scalability and efficiency to the end of the first paragraph of the "Comparison of variant scoring tools" section (page 16):

The computational resources required to execute each tool on a given dataset also vary due to implementation differences. DMS scoring tools use data frames to store and manipulate datasets, meaning that memory requirements scale linearly with the number of variants. For most workflows, runtime is dominated by FASTQ file processing (i.e., the process is I/O bound), since the files are large and the calculations performed to score each variant are not particularly intensive.

2. It may be helpful to give a sense of the workflow for each approach, especially for readers who are not well-versed in computational analyses. This has been described already in the text for most methods, but it could be helpful to make these descriptions more systematic, for example by incorporating how the software is run in Table 1 (e.g., GUI, notebook, command line interface).

We thank the reviewer for this excellent suggestion. We have added a new column to Table 1 called "Interface" that describes whether the tools are intended to be run using a CLI, GUI, or as a module imported by the user's own notebooks or scripts, along with a footnote defining these terms.

1st Jul 2025

Dear Dr Rubin,

Congratulations on an excellent review, I am pleased to inform you that your manuscript has been accepted for publication in Molecular Systems Biology.

Your manuscript will be processed for publication by EMBO Press. It will be copy edited and you will receive page proofs prior to publication. You will soon be contacted by Springer Nature to sign your publishing license. When you login to the customer service website, please use the following token to waive the article publication charges:
MTIWMZYXNTQONG

Should you experience any difficulty, please email publishing@embo.org.

If you have any other questions, please do not hesitate to contact the Editorial Office. Thank you for your contribution to Molecular Systems Biology.

Sincerely,

Poonam Bheda, PhD
Scientific Editor
Molecular Systems Biology